# Comparison of the Morphological Characteristics of South African Sub-Elite Female Football Players According to Playing Position

**DOI:** 10.3390/ijerph18073603

**Published:** 2021-03-31

**Authors:** Anita Strauss, Martinique Sparks, Cindy Pienaar

**Affiliations:** 1Department of Sport, Rehabilitation & Dental Science, Pretoria West Campus, Tshwane University of Technology, Pretoria 0001, South Africa; StraussA@tut.ac.za; 2Physical Activity, Sport and Recreation (PhASRec), North-West University, Potchefstroom 2520, South Africa; CindyP@dut.ac.za; 3Department of Sport Studies, Faculty of Applied Sciences, Durban University of Technology, Durban 4000, South Africa

**Keywords:** anthropometry, height, somatotype, body mass, soccer, sports performance

## Abstract

Limited information is available on the morphological characteristics of adult female footballers, therefore the aim of this article was to examine if there are position-specific differences in the morphological characteristics of sub-elite female football players and to establish normative standards for this level of female football players. The morphological features of 101 sub-elite female football players (age: 21.8 ± 2.7 years) were assessed. Twenty anthropometric sites were measured for body composition and somatotype. The average value of body fat percentage was 20.8 ± 5.7%. The somatotype of the overall group was 4.0–2.4–2.1. Significant (*p* ≤ 0.05) differences were found between goalkeepers and outfield players in morphological characteristics. Goalkeepers were taller (166.2 ± 8.4 cm), heavier (66.5 ± 5.1 kg), possessed the highest body fat percentage (17.2 ± 6.2%) and showed higher values for all skinfold (sum of 6 skinfolds = 125.6 ± 45.9 cm), breadth, girth and length measurements. However, there were very few practically worthwhile differences between the outfield positions. Positional groups did not differ (*p* ≤ 0.05) in somatotype characteristics either. The study suggests that at sub-elite level there are mainly differences between goalkeepers and outfield players, but outfield players are homogeneous when comparing morphological characteristics. These results may serve as normative values for future comparisons regarding the morphological characteristics of female football players.

## 1. Introduction

Female football is experiencing a constant growth in the number of players worldwide [1]. The efficiency of the game is influenced by several factors, including morphological characteristics, motor abilities, functional abilities, technical and tactical abilities [2]. The profile of football players has therefore become a subject of decisive interest [3]. Ample literature regarding the morphological characteristics of male football players exists, but less is known about female players [4]. To compete at an elite level, players need to possess morphological characteristics applicable to both the sport and specific playing positions [5].

A few studies have recently focused on the morphological characteristics of female football players participating at the elite and sub-elite level [6,7,8,9]. Results indicate that standing height and mass differ according to playing position in elite and sub-elite female football [7,10,11]. Elite goalkeepers are taller [10] and heavier [12], with defenders also being among the taller players [10]. Midfielders tend to be the shortest and lightest players, followed by forwards [13,14]. It is believed that a tall goalkeeper is advantageous in jumping to reach for a ball and defending a goal [15], while the shorter height and lighter body mass of midfielders enable them to move more efficiently and cover longer distances on the field [5]. Body fat percentage (BF%) also differs between playing positions in sub-elite female football, with goalkeepers having the highest body fat percentage [7]. This is possibly due to the lower level of activity required during a match [16].

Skeletal length and girth measurements show a general tendency to be larger for elite goalkeepers, with special reference to longer leg and arm lengths compared to other playing positions [13]. There is also a trend for midfielders to be among the players with the smallest skeletal length and girth measurements, which can again be attributed to their shorter stature, lower body mass and positional role [13]. The physical characteristics of the body gives an indication of somatotype [17]. Three somatotype components are used to define body shape; namely endomorph (relative fatness), mesomorph (musculoskeletal component) and ectomorph (linearity) and are expressed by a three-number rating [18]. Mesomorph is generally the most predominant somatotype component among football players [19], with an ectomorphic mesomorph body type being advantageous for actions requiring speed, power and endurance, which are vital components for football performance [20].

These above mentioned morphological characteristics are viewed as important considerations in the selection process for team positions [13]. Due to a shortage of related studies, a need exists for the establishment of normative data for female football players. The primary aim of this article is to examine if there are position-specific differences in the morphological characteristics of sub-elite female football players. A secondary aim is to utilise the morphological characteristics of sub-elite female football players to establish normative standards for this level of female football players.

## 2. Materials and Methods

### 2.1. Participants

The study comprised of 101 sub-elite female football players from five different clubs across the nine provinces of South Africa, with a mean age of 21.8 ± 2.7 years. Sub-elite level generally refers to players competing at club, college, or university level. In line with previous studies on female football players [12,14], the players were divided into four positional groups, namely forwards, midfielders, defenders and goalkeepers. All the participants were regular football players competing in local club-level and student tournaments with an average of 7.5 years playing experience. The participants practiced on a regular basis (2–3 times per week) and measurements were collected during the competition season (summer months). Participation in the study was voluntary and participants could withdraw from the study at any time without prejudice. Prior to the start of the study, the participants were duly informed of the purpose and experimental procedures and an explanation of the potential risks and benefits of the study were given. Each player received a participant number to ensure anonymity. The study was approved by the Health Research Ethics committee at the university where the research was conducted.

### 2.2. Anthropometric Measurements

Anthropometric measurements were taken according to the standard procedures of the International Society for the Advancement of Kinanthropometry [21]. Measurements included: (i) stature measured in centimeters to the nearest 0.1 cm using a Harpenden Portable Stadiometer (Holtain Limited, Crosswell, UK) with the player standing upright and the player’s head in the Frankfort plane; (ii) body mass measured in kilograms to the nearest 0.1 kg using a portable electronic scale (Ps07 Electronic Scale, Beurer, Ulm, Germany) with the participants wearing minimal clothing (such as shorts and a crop top) and no shoes; (iii) skinfolds of the triceps, subscapular, supraspinal, abdominal, frontal thigh and medial calf were measured with a Harpenden Skinfold Caliper (Holtain Limited) to the nearest 0.2 mm with a constant pressure of 10 g/mm^2^; (iv) breadths of the humerus, wrist, femur and ankle were measured with a Holtain Bicondylar Caliper (Holtain Limited) to the nearest 0.1 mm; (v) girth measurements for relaxed arm, flexed arm, waist, gluteus and mid-thigh taken with a Lufkin metal tape (Cooper Industries, Cleveland, OH, USA) to the nearest 0.1 cm; and (vi) skeletal lengths of the upper arm, lower arm, hand and foot were measured with a Rosscraft segmometer (Rosscraft Innovations Incorporated, Granville, YVR, Canada) to the nearest 0.1 cm. All anthropometric measurements were taken by the same two Level 2 ISAK-certified anthropometrists twice on the right-hand side of the body. The mean values of these measurements were used in the statistical analysis. Body fat percentage [22] and muscle and skeletal mass [23] as well as somatotype [18] was calculated using previously established formulas. For the full description of measurements and formulas used please find the Appendix A attached to this article. Arm, mid-thigh and calf girths were corrected with the different skinfolds at these sites by applying the following formula: Corrected arm girth = Girth − (3.1416 × (triceps skinfold/10)); Corrected thigh girth = Girth − (3.1416 × (thigh skinfold/10)); Corrected calf girth = Girth − (3.1416 × (calf skinfold/10)). Corrected girths were used because they serve as better indicators of musculoskeletal size at each site [24]. The technical error of measurement was calculated according to the method of Pederson and Gore [25]. A value of 1.80% (1.75 mm) was revealed for all skinfold measurements, 0.62% (0.16 cm) for all breadth measurements, 0.01% (0.02 cm) for all girth measurements and 0.87% (0.19 cm) for all length measurements. Thus, indicating that all anthropometric measurements were reliable for the purpose of this study.

### 2.3. Statistical Analyses

The Statistical Package for Social Science (SPSS Statistics 26, IBM, Armonk, NY, USA) was used for statistical analysis. Descriptive statistics were calculated and used to describe the morphological characteristics of the players.

### 2.4. Positional Differences

The statistics were reported as mean ± standard deviation (SD). One-way analysis of variance (ANOVA) was used to determine differences between the morphological characteristics among the four playing positions. Scheffe’s F test was also used for multiple comparisons between groups. Tukey’s post hoc test was used to determine which variables differed significantly. The level of statistical significance was set a *p* ≤ 0.05.

Due to the sampling variability, effect size (Cohen’s D) and 90% confidence intervals were used to compare differences in standardised effects between playing positions. Magnitudes of standardised effects were assessed as: 0–0.2 trivial, 0.2–0.6 small, 0.6–1.2 moderate, 1.2–2.0 large, and >2.0 very large [26]. An effect size greater than 0.2 was seen as a worthwhile change, however if the lower and upper confidence intervals exceeded −0.2 and 0.2 the difference is deemed unclear and no inference can be made on whether “true” differences can be observed in the greater population [26].

### 2.5. Normative Classification

For the purpose of normative classification, the body composition variables were classified using standard nine (“stanine”) scores which scaled the parameters from “extremely low” to “extremely high” [27].

## 3. Results

Descriptive statistics of the morphological characteristics for the total group and differences according to specific playing positions are presented in Table 1. The players had a mean standing height and mass of 160.0 cm and 57.1 kg respectively. Goalkeepers were statistically taller than midfielders (166.2 vs. 158.7 cm) and defenders (166.2 vs. 159.1 cm) as well as heavier (66.5 kg) than the forwards (56.3 kg), midfielders (55.0 kg) and defenders (57.4 kg). With regards to statistical significance, the only differences were found between the goalkeepers and the outfield players. No differences were found between forwards, midfielders, and defenders.

All positional groups had average endomorphic values (Table 2). While the forwards and midfielders presented a balanced endomorph somatotype, the defenders and goalkeepers showed an average mesomorphic-endomorph somatotype. Descriptive statistics for body composition characteristics and comparisons between playing positions are presented in Table 2. Once again, the only statistically significant differences were between goalkeepers and the outfield players. However, from Figure 1 it is apparent that although some effect sizes were larger than 0.2 for outfield position differences, the upper and lower confidence intervals were beyond 0.2 and −0.2, which means that the inference is deemed unclear. The only clear inferences that can be made is for differences between goalkeepers and outfield players. However, forwards had longer Radiale-stylion and Acrimiale-radiale lengths when compared to midfielders and broader ankle and humerus breadths than defenders. On the other hand, defenders had broader ankle, femur and humerus breadths when compared to midfielders. For most measurements, goalkeepers had larger skinfolds, girths, circumferences, and longer bone lengths. Goalkeepers also had a lower relative muscle mass (Effect size > 0.6) and a higher fat % (Effect size > 0.8). Due to no differences between the outfield players, each of the body composition components are presented according to stanine categorizations in Table 3 to provide a method of classifying each component from “extremely low” to “extremely high”. For example, an ‘above average’ fat % for an outfield player would be between 20.8% and 24.4%. No stanines are provided for goalkeepers due to the small sample size (*n* = 9).

## 4. Discussion

The primary aim of this article was to examine if there are position-specific differences in the morphological characteristics of South African sub-elite female football players. A secondary aim was to utilise the morphological characteristics of South African sub-elite female football players to establish normative standards for this level of female football players. Adding to the uniqueness of this study is the large sample size in comparison to most other studies on female football players, cited in this paper. The main finding of this study was that the biggest differences (statistically and practically) were between goalkeepers and outfield players, with limited differences between the outfield positions.

Height and body mass values in the current study were found to be comparable with values reported for elite Spanish [4] and sub-elite Japanese [7] female football players. However, most previous research on female football players reported considerably higher values for standing height and body mass among both sub-elite [9,14] and elite female players [10,11]. Height is regarded as a decisive factor in the selection process in football and is considered favorable for goalkeepers when defending a goal [13]. Similar to other studies [10,14,28], goalkeepers were taller than the outfield players, with results being significantly different (*p* < 0.02) compared to midfielders and defenders. None of the outfield playing positions differed (*p* > 0.05) from one another in terms of height. However, like previous studies [3,4,13,14], midfielders were the shortest players, whilst the height of forwards and defenders were alike. It would be expected that the height of forwards would be similar to defenders, due to the direct influence it would have in duels when jumping in front of the goalposts [29]. The height values were partly accompanied by the higher total body mass values, leading to goalkeepers being the heaviest and midfielders the lightest, thus supporting previous findings in the literature [12,13]. The heavier body mass often observed in goalkeepers is presumably because endurance is less important for goalkeepers than for outfield players [15]. Various skeletal measurements should also be considered. Skeletal breadth, girth, length and skinfold measurements of the players in the current study were comparable with other female football players participating at the sub-elite and elite level, although available literature is very limited for all levels of participation [1,4,13,20,30]. Results from the current study were either similar (thigh girth and humerus, wrist, femur and ankle breadth) or lower (relaxed arm, flexed arm and waist and calf girth) than previous reports [1,13,31]. Upper limb skeletal length for players were longer than those in previous reports but cannot be directly related since this study made use of three measurements (mean acromiale-radiale, mean radiale-stylion and mean midstylion-dactylion) in comparison to the single measurement used for arm length in previous reports [1,13,31].

Differences (*p* ≤ 0.05) were found between playing positions in the mean breadth measurements of the wrist, femur and ankle and the mean girth measurements of the relaxed arm, flexed arm, waist, gluteus and medial calf. Goalkeepers had larger (*p* ≤ 0.05) wrist breadth measurements compared to midfielders and defenders and larger femur breadth measurements compared to forwards (*p* = 0.02) and midfielders (*p* = 0.003). A previous study on female players also reported the largest breadth measurements among goalkeepers [13]. The forwards had the smallest humerus and ankle breadth measurements, followed by the midfielders who again had the smallest wrist and femur breadth measurements. In terms of girth measurements, differences (*p* ≤ 0.05) were found between goalkeepers and the other three positional groups, with the largest measurements being noted among goalkeepers. Midfielders possessed the smallest mean results for the majority of girths measured, which concurs with previous studies [13]. Due to limited data being available on female players, comparisons to other studies are problematic, but from these results it is clear that goalkeepers have wider bone girths compared to the outfield players.

Players in different positions further varied (*p* ≤ 0.05) in terms of their mean length measurements. Goalkeepers had longer (*p* ≤ 0.05) acromiale-radiale length measurements than midfielders and defenders and longer (*p* ≤ 0.01) radiale-stylion length measurements compared to all outfield positions, thus agreeing with previous reports [13]. This particular body type contributes to the self-confidence of goalkeepers in performing their tasks of covering the broad area between the goalposts [29]. Once again, the midfielders had the shortest arm and hand lengths, as previously reported [13]. Furthermore, goalkeepers had larger hands (*p* ≤ 0.01) and longer feet (*p* ≤ 0.05) than midfielders and defenders. In considering standing height and body mass values of the different playing positions, it was not unexpected that the shorter and lighter midfielders generally possessed the smallest skeletal breadth, girth and length measurements. In addition the largest measurements, accompanying the tallest and heaviest values, were established among the goalkeepers.

Skinfold and body fat measurements are major contributing factors for determining a player’s body composition. Although a certain amount of body fat is important for the maintenance of body metabolism, it is believed that excess adiposity negatively influences football performance [32,33]. The sum of the skinfolds can be used to determine adiposity and to provide detail regarding local fat depots and fat distribution in the body [34]. The average of the sum of six skinfolds was considerably higher than that previously reported [30] for elite female players and was more in agreement with the values reported for junior female players. No statistical difference was found between the different playing positions, however effect sizes indicated again that large differences existed between goalkeepers and outfield players, with goalkeepers having higher values in the sum of skinfolds. The body fat mass and mean BF% of the players were within the ranges previously reported for female players [4,7,9]. Higher muscle mass values than reported in this study were observed for sub-elite and elite Spanish female players [4]. Differences were noted between the different playing positions concerning the muscle mass (*p* < 0.01), skeletal mass (*p* < 0.01) and fat mass (*p* < 0.05) values when goalkeepers were compared to the outfield positions. No differences were observed when the outfield positions were compared with one another. The higher BF% values found among goalkeepers could be attributed to their specific positional requirements of being steadier throughout the game and less mobile. This is in comparison to other playing positions that require the players to be more maneuverable and to cover a greater distance throughout a match, which could explain the lower BF% measured among them. Forwards and defenders, however, had similar BF% values compared to midfielders, indicating that BF% was not a distinguishing factor among outfield positions. This agrees with results previously reported for elite Japanese female players [7] and elite and sub-elite Spanish [4] female players.

Finally, somatotype is a useful unit of measure that highlights the overall health status of individuals [35]. In football, the mesomorphic component (strength indicator) together with a prevalence of ectomorphic components is considered optimal for performance [20]. Therefore, an ectomorphic mesomorph body type is more desirable for performance in football as the sport requires speed, endurance and muscle power [20]. Although there is a shortage of studies describing the somatotype of female football players, the majority reported an endomorphic mesomorph body type [20,22,31], with one study reporting a mesomorphic endomorph body type [9]. Table 4 contains a summary of studies that investigated differences in in somatotype and fat percentage in female football players. The players in this study demonstrated a balanced endomorph body type. This can be translated into an average value for the endomorphic component and low values for both the mesomorphic and ectomorphic components, indicating that on average, the players had a shorter body type and less muscular characteristics compared to previously reported results. None of the positional groups differed in somatotype characteristics. Comparisons to other studies regarding somatotype characteristics according to playing position could, however, not be made since such analyses have not been conducted previously.

### 4.1. Limitations

Although this study provides insight to the morphological characteristics of sub-elite female football players, the small number of goalkeepers included in this study is a limitation. Even though most teams only include two goalkeepers as part of their squad during tournaments, future studies should attempt to recruit additional goalkeepers to increase their sample size.

### 4.2. Strengths

Our study included a large sample (*n* = 101) of top semi-elite football players. Apart from the study by Tod et al. [38] (*n* = 120) our sample size was considerably more than previous studies on adult female football players, and therefore the results can be used by coaches as a refence for the body composition of their players.

### 4.3. Practical Implications

This study provides normative data for coaches to use as a reference for the body composition of their players. By comparing players to the normative data coaches can be guided on how to adapt conditioning programs to elicit improvement in body composition and performance.

## 5. Conclusions

This study provides valuable information concerning the morphological profile of sub-elite female football players in accordance with playing position. The results of the present study suggest that goalkeepers differ in morphological characteristics compared to outfield players, but the outfield players were homogeneous when compared to each other. The results support previous research in that players in defensive positions tend to be among the taller players, which could be advantageous in performing actions such as jumping to gain possession of the ball and defending the goalpost. Midfielders are likely to be the shortest and lightest players in a team, lending them the advantage of being more maneuverable across the field. The inclination of the endomorphic component may be viewed as an indication of undertraining. Finally, the stanine provided for outfield players provide guidelines for body composition and contribute towards normative data within an area characterized by a rarity of information.

## Figures and Tables

**Figure 1 ijerph-18-03603-f001:**
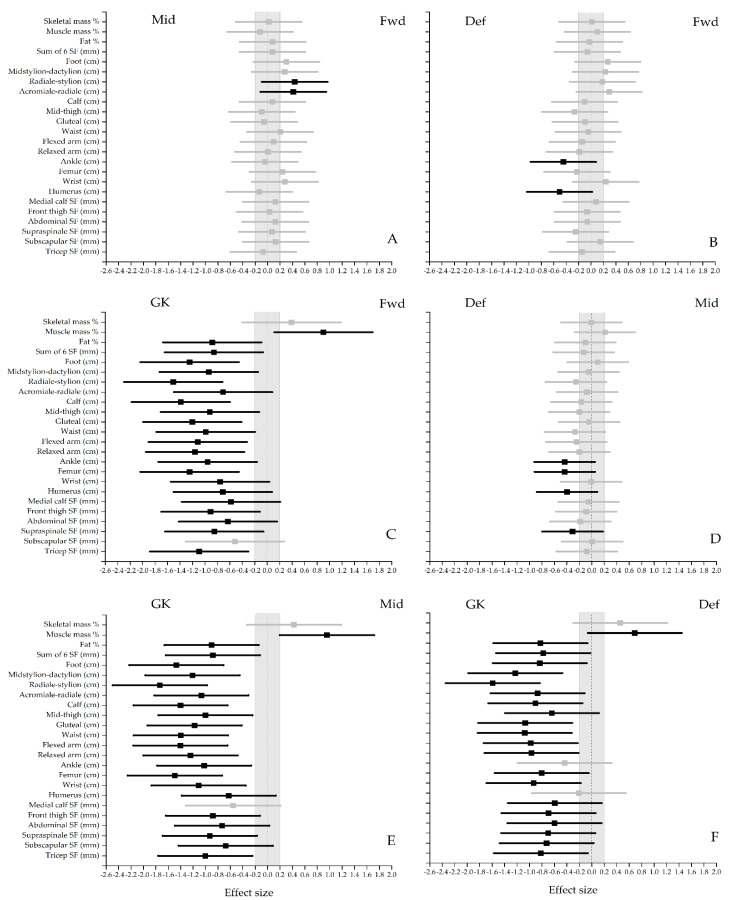
Effect size for position-specific anthropometric characteristics (95% CI). Differences between: Midfielders vs. Forwards (**A**); Defenders vs. Forwards (**B**); Goalkeepers vs. Forwards (**C**); Defenders vs. Midfielders (**D**); Goalkeepers vs. Midfielders (**E**); Goalkeepers vs. Defenders (**F**). Points estimates are colour-coded as “unclear” (grey) and “meaningful” (black). Negative effect sizeds indicate a higher measurement for the position named first in the description.

**Table 1 ijerph-18-03603-t001:** Morphological characteristics of sub-elite female football players by playing position (mean ± SD).

Variables	Total (*n* = 101)	FW (*n* = 25)	MF (*n* = 33)	DF (*n* = 34)	GK (*n* = 9)	*F*-Value	*p*-Value
Body stature (cm)	160.0 ± 6.8	160.9 ± 5.7	158.7 ± 6.1 *	159.1 ± 6.9 *	166.2 ± 8.4	3.5	0.018
Body mass (kg)	57.1 ± 9.1	56.3 ± 8.4 *	55.0 ± 8.4 *	57.4 ± 9.9 *	66.5 ± 5.1	4.2	0.008
**Skinfolds (mm)**							
Tricep	16.0 ± 5.3	15.1 ± 4.7 *	15.5 ± 4.9 *	15.9 ± 5.7	20.5 ± 5.8	2.7	0.052
Subscapular	11.4 ± 4.7	11.4 ± 4.5	10.9 ± 3.9	10.9 ± 3.4	14.6 ± 9.4	1.7	0.169
Supraspinale	10.3 ± 5.3	9.6 ± 4.1 *	9.3 ± 4.4 *	10.7 ± 4.5	15.0 ± 10.4	3.1	0.030
Abdominal	19.2 ± 8.3	18.8 ± 8.0	17.8 ± 8.4	19.3 ± 7.7	24.2 ± 9.9	1.4	0.233
Front thigh	24.9 ± 10.2	24.1 ± 8.9	23.8 ± 9.7	24.7 ± 11.3	32.4 ± 9.9	1.8	0.145
Medial calf	15.7 ± 6.1	15.9 ± 5.1	15.1 ± 7.2	15.4 ± 5.9	18.9 ±5.3	0.9	0.421
**Breadths (cm)**							
Humerus	6.1 ± 0.3	6.0 ± 0.3	6.1 ± 0.3	6.2 ± 0.4	6.3 ± 0.5	2.2	0.097
Wrist	5.1 ± 0.3	5.1 ± 0.3	5.1 ± 0.2 *	5.1 ± 0.3 *	5.3 ± 0.3	2.8	0.044
Femur	8.9 ± 0.5	8.9 ± 0.4 *	8.7 ± 0.4 *	9.0 ± 0.6	9.4 ± 0.5	4.7	0.004
Ankle	6.5 ± 0.4	6.4 ± 0.4	6.4 ± 0.3	6.6 ± 0.4	6.7 ± 0.3	3.0	0.034
**Girths (cm)**							
Relaxed arm	24.9 ± 2.6	24.5 ± 2.6 *	24.5 ± 2.4 *	25.0 ± 2.5	27.4 ± 2.1	3.6	0.017
Flexed arm	26.4 ± 2.4	26.1 ± 2.5 *	25.9 ± 2.1 *	26.4 ± 2.4 *	28.7 ± 1.9	3.8	0.012
Waist	68.0 ± 5.9	67.7 ± 6.5 *	66.5 ± 5.3 *	68.0 ± 5.5 *	73.9 ± 5.4	4.0	0.010
Gluteal	93.4 ± 6.7	92.4 ± 6.7 *	92.8 ± 6.4 *	93.1 ± 6.7 *	99.9 ± 4.4	3.3	0.022
Mid-thigh	51.2 ± 4.6	50.3 ± 4.9	50.7 ± 3.9	51.6 ± 4.9	54.6 ± 4.0	2.3	0.085
Calf	32.9 ± 2.6	32.6 ± 2.1 *	32.4 ± 2.2 *	32.9 ± 3.0 *	35.5 ± 2.0	3.7	0.014
**Lengths (cm)**							
Acromiale-radiale	30.7 ± 1.6	31.0 ± 1.3	30.4 ± 1.4 *	30.5 ± 1.7	32.0 ± 1.9	3.0	0.032
Radiale-stylion	24.2 ± 1.5	24.3 ± 1.2 *	23.8 ± 1.4 *	24.1 ± 1.3 *	26.3 ± 1.6	8.6	0.000
Midstylion-dactylion	18.5 ± 1.0	18.5 ± 1.0	18.3 ± 1.0 *	18.3 ± 0.9 *	19.5 ± 1.2	4.2	0.008
Foot	24.4 ± 1.6	24.6 ± 1.0	24.3 ± 1.1	24.1 ± 2.2 *	25.8 ± 0.7	3.0	0.033

FW: Forwards; MF: Midfielders; DF: Defenders; GK: Goalkeepers; * Differs significantly from goalkeepers at *p* ≤ 0.05.

**Table 2 ijerph-18-03603-t002:** Body composition characteristics of sub-elite female football players by positional group (mean ± SD).

Variables	Total (*n* = 101)	FW (*n* = 25)	MF (*n* = 33)	DF (*n* = 34)	GK (*n* = 9)	*F*-Value	*p*-Value
Sum of 6 skinfolds (mm)	97.5 ± 35.8	94.9 ± 31.7	92.3 ± 35.4	96.9 ± 34.2	125.6 ± 45.9	2.2	0.092
Fat (kg)	20.8 ± 5.7	20.4 ± 5.1 *	20.0 ± 5.7 *	20.6 ± 5.5 *	25.4 ± 7.2	2.3	0.081
Fat %	12.2 ± 5.2	11.8 ± 4.6	11.3 ± 4.8	12.2 ±5.1	17.2 ± 6.2	2.4	0.022
Muscle mass (kg)	21.3 ± 2.1	21.1 ± 1.9 *	20.8 ± 1.9 *	21.3 ± 2.2 *	23.7 ± 1.0	5.2	0.002
Muscle mass %	37.7 ± 2.7	37.9 ± 2.4	38.2 ± 2.6	37.6 ± 2.8	35.7 ± 2.0	2.1	0.107
Skeletal mass (kg)	6.8 ± 0.9	6.8 ± 0.7 *	6.6 ± 0.7 *	6.9 ± 1.0 *	7.7 ± 1.0	4.3	0.007
Skeletal mass %	12.1 ± 1.2	12.1 ± 1.4	12.1 ± 1.2	12.1 ± 1.1	11.6 ± 1.2	0.5	0.693
Endomorphy	4.0 ± 1.3	3.8 ± 1.2	3.8 ± 1.2	4.0 ±1.3	4.9 ± 1.8	1.8	0.147
Mesomorphy	2.4 ± 0.9	2.0 ± 0.9	2.3 ± 0.7	2.7 ± 1.0	2.5 ±1.2	1.8	0.145
Ectomorphy	2.1 ± 1.2	2.4 ± 1.4	2.1 ± 1.0	1.9 ± 1.3	1.7 ±1.3	1.2	0.374

FW: Forwards; MF: Midfielders; DF: Defenders; GK: Goalkeepers * Differs significantly from goalkeepers at *p* ≤ 0.05.

**Table 3 ijerph-18-03603-t003:** Stanines for the body composition of outfield female football players.

	Low					Average						High				
	Extremely low	Very Low		Low		Below average		Average		Above average		High		Very high		Extremely high
Sum of 6 skinfolds (mm)	46.5		58.9		67.2		77.4		98.1		121.2		145.8		163.2	
Fat (%)	12.9		14.5		16.1		17.6		20.8		24.4		28.7		31.6	
Fat (kg)	6.2		7.1		8.1		9.2		11.5		15.1		18.4		23.6	
Muscle mass (kg)	18.2		18.9		19.4		20.6		21.3		22.1		23.7		26.1	
Muscle mass%	32.8		34.4		36.1		37.7		38.6		39.9		41.3		42.5	
Skeletal mass (kg)	5.5		5.8		6.2		6.4		6.8		7.4		7.7		8.4	
Skeletal mass (%)	9.8		10.7		11.0		11.9		12.5		13.1		13.8		14.3	

**Table 4 ijerph-18-03603-t004:** Summary table of studies examining differences between position in age, height, weight, %BF and somatotype of adult female football players (mean ± SD).

Author	Country & Competition Level	*n*	Age (Years)	Height (cm)	Weight (kg)	Body Fat (%)	Endomorphy	Mesomorphy	Ectomorphy
Nikolaidis [9]	Greek Amateur club level	54					5.2 ± 1.5	4.6 ± 1.3	2.0 ± 1.2
Milanovic et al. [10]	Serbian A-National Team	22	23.9 ± 4.5	168.8 ± 7.2 164.7 ± 5 (FW) 168.7 ± 8.7 (MF) 170.0 ± 7.2 (DF) 172.5 ± 3.5 (GK)	61.4 ± 6	25.9			
					62.7 ± 7.7 (MF)				
					59.5 ± 10.6 (GK)				
Sporis et al. [11]	Croatian elite level total group	24	18.3 ± 0.6	165.6 ± 4.2	58.3 ± 4.6	21.3 ± 1.5			
		4	17.4 ± 0.4 (FW)	165.0 ± 4.2 (FW)	63.6 ± 4.1 (FW)	20.3 ± 1.7 (FW)			
		12	18.3 ± 0.7 (MF)	164.0 ± 4.3(MF)	56.0 ± 4.8 (MF)	21.6 ± 1.8 (MF)			
		5	18.5 ± 0.6 (DF)	165.8 ± 3.9 (DF)	56.3 ± 4.9 (DF)	21.8 ± 0.9 (DF)			
		3	19.1 ± 0.5 (GK)	172.5 ± 4.6 (GK)	64.4 ± 4.2 (GK)	20.7 ± 1.2 (GK)			
Haugen et al. [12]	Norwegian National Team	85	23.5 ± 3.6		63.7 ± 5.2				
	Norwegian 1st Division	46	21.2 ± 3.6		62.4 ± 6.6				
	Total group	44	21.9 ± 3.8 (FW)		64.1 ± 6.7 (FW)				
		55	21.6 ± 4.3 (MF)		61.5 ± 4.6 (MF)				
		50	21.6 ± 4.1 (DF)		61.9 ± 5.7 (DF)				
		16	21.4 ± 4.7 (GK)		67.3 ± 4.6 (GK)				
Sporis et al. [13]	Croatian First League club players	24		165.6 ± 5.8	58.6 ± 9	13.6 ± 4.2			
				165.0 (FW)	63.6 (FW)	14.3 (FW)			
				164.0 (MF)	56.0 (MF)	12.6 (MF)			
				165.8 (DF)	56.3 (DF)	16.8 (DF)			
				172.5 (GK)	64.4 (GK)	13.7 (GK)			
Vescovi et al. [14]	USA Sub-elite 1st Division	64	19.8 ± 1.2	168.4 ± 5.9	64.8 ± 5.9				
				168.3 ± 6.6 (FW)	64.5 ± 5.8 (FW)				
				165.9 ± 6.3 (MF)	61.3 ± 4.7 (MF)				
				169.9 ± 4.3 (DF)	67 ± 6.7 (DF)				
				170.3 ± 5.7 (GK)	66.4 ± 1.9 (GK)				
Adhikari and Nugent [20]	Canadian club level	18					3.0 ± 0.8	3.9 ± 0.8	2.6 ± 1.0
Can et al. [31]	Turkish highest division	17					3.0 ± 0.8	3.5 ± 0.9	2.4 ± 0.9
Withers et al. [22]	South Australian team	11					4.2 ± 1.3	4.6 ± 1.0	2.2 ± 1.2
Goranovic et al. [36]	Serbian super league	20	19.7 ± 4.8	163.8 ± 3.7 (FW)	56.3 ± 5.8 (FW)	20.9 ± 4.2 (FW)	2.71 (FW)	3.46 (FW)	2.74 (FW)
				167.1 ± 7.5 (MF)	61.3 ± 7.4 (MF)	22.8 ± 4.6 (MF)	3.32 (MF)	3.43 (MF)	2.53 (MF)
				171.2 ± 6.1 (DF)	61.7 ± 6.8 (DF)	20.8 ± 5.0 (DF)	2.89 (DF)	2.87 (DF)	3.19 (DF)
Krustrup et al. [37]	Danish highest division	14	24	167.0	58.5	14.6			
				166.0 ± 4 (FW)	58.7 ± 3.8 (FW)	16.1 ± 2.4 (FW)			
				165.0 ± 4 (MF)	56.0 ± 5.9 (MF)	12.5 ± 2.2 (MF)			
				168.0 ± 7 (DF)	60.7 ± 6.3 (DF)	15.4 ± 3.7 (DF)			
Todd et al. [38]	English international	25	22.3 ± 4.3 23.4 ± 5.9 21.3 ± 6.6	162.8 ± 5.9	61.2 ± 5.2	22.9 ± 3.4			
	English Premier League	44		163.3 ± 5.5	62.1 ± 6.4	23.9 ± 4.2			
	English Regional League	51		163.9 ± 6.3	61.6 ± 7.1	25.5 ± 3.5			
	Total group	120		162.5 ± 6.8 (FW)	60.9 ± 7.3 (FW)	24.3 ± 4.1 (FW)			
				161.6 ± 5 (MF)	59.5 ± 5 (MF)	24.0 ± 3.5 (MF)			
				165.2 ± 5.6 (DF)	62.7 ± 6.6 (DF)	24.2 ± 3.9 (DF)			
				168.5 ± 4.3 (GK)	68.9 ± 5.5 (GK)	26.3 ± 4.3 (GK)			
This study	South African semi-elite	101	21.8 ± 2.7	160.9 ± 5.7 (FW)	56.3 ± 8.4 (FW)	11.8 ± 4.6 (FW)	3.8 ± 1.2 (FW)	2.0 ± 0.9 (FW)	2.1 ± 1.2 (FW)
				158.7 ± 6.1 (MF)	55.0 ± 8.4 (MF)	11.3 ± 4.8 (MF)	3.8 ± 1.2 (MF)	2.3 ± 0.7 (MF)	2.1 ± 1.0 (MF)
				159.1 ± 6.9 (DF)	57.4 ± 9.9 (DF)	12.2 ± 5.1 (DF)	4.0 ± 1.3 (DF)	2.7 ± 1.0 (DF)	1.9 ± 1.3 (DF)
				166.2 ± 8.4 (GK)	66.5 ± 5.1 (GK)	17.2 ± 6.2 (GK)	4.9 ± 1.8 (GK)	2.5 ± 1.2 (GK)	1.7 ± 1.3 (GK)

FW: Forwards; MF: Midfielders; WM: Wide Midfielders; DF: Defenders; CB: Centre Back; FB: Full Back; GK: Goalkeepers.

## Data Availability

According to the approval from the Health Research Ethics committees, the data are to be stored properly and in line with the South African Law of privacy protection. Public availability would compromise privacy of the respondents. However, anonymized data are available to interested researchers upon request, pending ethical approval from our Ethics committee. Interested researchers can contact project leader Martinique Sparks (martinique.sparks@nwu.ac.za) with requests for the data underlying our findings.

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
