# Peer review of "Comparison of the Morphological Characteristics of South African Sub-Elite Female Football Players According to Playing Position"

_ijerph, 2021, doi:10.3390/ijerph18073603_

Round 1

Reviewer 1 Report

1.    Abstract – add n size and some data. Is this a model study to look for profiles of other players physically to match this? Confusing as written.

2.    IF it is a profile study how does it related to other studies in the literature on soccer players? You could use a table with your analytics and others to see how they match up and with ranges etc. See work in the IJERPH on anthropometry and somatotype and do a more complete literature search for data similar to yours in the MDPI.

3.    I would expect somatotype would be more mesomorphic for good spike velocity. Also, is this information really necessary even if this study assess a sub-elite sample? what is trainable or not is the important key and then what they do to train the trainable metrics. Or it is just descriptive and not much help.

4.    What is the training background coming into the study and what time of year etc were they tested.

5.    The statistical analysis needs to be in context of how the data were analyzed in respect to the hypotheses to be tested. Low n size and thus were the statistical assumptions met and if not what was done then? What was the test retest reliability e.g., ICCRs SEM and what are your confidence intervals. Profiles have to be useful and I am not sure how to use your data? Did you correct for inflating alpha level with repeated use of T tests on multiple parameters?

6.    Results: Each paragraph should be logical in sequence as at present it is a bit hard to follow.

7.    I find this paper of little use if in fact it is of former players not in training and detrained, how long and training background is important when tested as we need to know the mutable aspects of the training. I work with club and elite soccer players and how does this paper help me as to my use of it in training these athletes for target aspects you present. I do not get it. Unless this is clear it makes no sense to target parameters that are of a detrained nature etc.

8.    In general, the discussion section is very descriptive and offers limited comparisons to previous research. In this regard, the MDPI literature should be consulted as recently innovative studies have been proposed regarding the variables considered. Even if they are not about female athletes, here are some useful titles published in MDPI journals:

-    doi: 10.3390/ijerph17218176. Prediction of Somatotype from Bioimpedance Analysis in Elite Youth Soccer Players

-    doi: 10.3390/ijerph16245066. Classic Bioelectrical Impedance Vector Reference Values for Assessing Body Composition in Male and Female Athletes

-    https://doi.org/10.3390/su12176789. Optimal Body Composition and Anthropometric Profile of World-Class Beach Handball Players by Playing Positions

Author Response

Thank you for all of your constructive comments. Please find attached the reply

Reviewer 2 Report

I would like to thank the editorial board for the opportunity to review the current manuscript that assessed the “Comparison of the morphological characteristics of sub-elite female football players according to playing position”. The current paper makes a valuable contribution to the literature concerning female football players. There are some minor comments that I would like to see addressed.

Introduction

L38 - ....participating on elite and sub-elite level.... replace with participating at the elite and sub-elite level

There was no mention of the differences in elite and sub-elite until the aim of the study. Please include details of the differences between the players at each level. This is important as players at the sub-elite level may not be participating in the same volume of training and level of competition.

Methods

You mentioned the formulas that you used to calculate the value for each site. Based on the results of each player, please add details about the method used to place the players in a somatotype category

Results

You mention on numerous occasions “significant” differences. You may remove the word significant as when you mention that the positions were different it is generally accepted that the results were significant

Tables

Table 1 – using a * to show significance between positions for all positions is not clear. In the current form, it is difficult to interpret which positions are different from each other. Please use different letters or symbols for each position. This helps the reader to identify the differences between positions. I noticed in table 2 that you used * different than goalkeepers, if this is the same in table 1, please make it clearer for the reader

Table 2 – can you check the row concerning Fat (kg). The total (n = 101) column shows the mean was 12.2 ± 5.2, whereas the mean in each of the positions were in the 20’s.

Discussion

L167 – ....body mass values of the current study.... replace with ....body mass values in the current study....

L182 – ...findings in literature... replace with ...findings in the literature...

L186 - .....players on elite and sub-elite level.... replace with participating at the elite and sub-elite level

Author Response

(The authors gave the same response as above.)

Reviewer 3 Report

The introduction seems well framed, and yet there are typographical errors that with a quick reading could be solved

The materials and methods are presented in a complex way to the reader. I recommend that these be segmented, especially in the section referring to statistical treatment, assigning a subsection for this.

The results section is incomplete. I recommend that you divide and develop the results according to tables / figures, point by point, rather than a summary.

The conclusions section is also incomplete, it requires adding study limitations and practical applications.

It would be interesting to see a comparative study based on gender as a future line of research by the authors.

Author Response

(The authors gave the same response as above.)

Reviewer 4 Report

The work entitled "Comparison of the morphological characteristics of sub-elite female football players according to playing position " has an interesting approach for publication in Int. J. Environ. Res. It is very interesting to know the body characteristics of the players to improve sports performance. But, there are some questions of form that should be taken into account prior to consider this article for publication.

I attach the coments to author with the changes.

  • the title must reflect the population studied
  • The abstract does not include all parts of the work. I believe that it should be reformulated, the authors should include the contextualization and justification of the work. they should also reflect a more concrete conclusion.
  • Methods:

-The authors must reflect the number of sports clubs that participated, and if they belong to the same región.

-Despite being reflected in another section of the manuscript, I consider that this tab section should reflect the approval of the ethics committee.

-I consider it important to reflect in more detail the anthropometric measurement protocol; What reference point was taken to make each measurement? (line92)

  • The discussion should deal with the following topics:

-In the age group studied, we can consider that some participants would still be finishing their period of physiological development or growth. So, this could be considered a limitation of the study? How might this have affected the results?

-The results are compared with the data of other works in a general way. We know that different ethnic groups have their own morphologies. Therefore, the discussion should focus on more specific results, that is, there are the same differences between the different types of Spanish and Japanese players with respect to the population studied.

  • The conclusions must be formulated, they must specifically answer the objective.

Author Response

(The authors gave the same response as above.)

Round 2

Reviewer 4 Report

The authors resolved most of the suggestions proposed in the previous review at work. Despite this, I still consider that the authors should include in the title that the subjects studied are from South Africa.

Author Response

We have changed the title to: Comparison of the morphological characteristics of South African sub-elite female football players according to playing position